# Examining the Impact of Two Dimensions of Precarious Employment, Vulnerability and Insecurity on the Self-Reported Health of Men, Women and Migrants in Australia

**DOI:** 10.3390/ijerph17207540

**Published:** 2020-10-16

**Authors:** Alison Daly, Marc B. Schenker, Elena Ronda-Perez, Alison Reid

**Affiliations:** 1School of Public Health, Faculty of Health Sciences, Curtin University, Kent Street, Bentley, Western Australia 6102, Australia; alison.daly@curtin.edu.au; 2Department of Public Health Sciences, School of Medicine, University of California, Davis, CA 95616, USA; mbschenker@ucdavis.edu; 3Public Health Research Group, Faculty of Health Sciences, University of Alicante, Carretera San Vicente del Raspeig s/n, 03690 Alicante, Spain; elena.ronda@ua.es

**Keywords:** precarious employment, migrant workers, cross-sectional, self-reported health

## Abstract

Precarious employment is increasing and adversely affects health. We aimed to investigate how perception of precariousness in current employment impacts gender and migrant workers in Australia. Using cross-sectional interviews of 1292 workers born in Australia, New Zealand, India and the Philippines, data were collected on self-reported health, employment conditions and sociodemographics. Factor analysis of nine questions about perceptions of current employment revealed two dimensions, vulnerability and insecurity. Women had higher vulnerability scores (µ = 6.5 vs. µ = 5.5, t = 5.40, *p*-value (*p*) < 0.000) but lower insecurity scores (µ = 8.6 vs. µ = 9.3 t = −4.160 *p* < 0.0003) than men. Filipino-born workers had higher vulnerability compared with other migrant workers (µ = 6.5 vs. µ = 5.8 t = −3.47 *p* < 0.0003), and workers born in India had higher insecurity compared with other migrant workers (µ = 9.8 vs. µ = 8.9, t = −6.1 *p* < 0.0001). While the prevalence of insecurity varied by migrant status, the negative effect on health was higher for Australian-born workers than migrants. Increasing levels of vulnerability and insecurity impacted self-reported health negatively (Coefficient (Coef).0.34 *p* < 0.0001; Coef.0.25 *p* < 0.0001, respectively). The combination of high vulnerability and high insecurity had the greatest impact on health (Coef. 2.37 *p* = 0.002), followed by high vulnerability and moderate insecurity (Coef. 2.0 *p* = 0.007). Our study suggests that understanding both changes in employment conditions over time as well as knowledge of cultural patterns may offer the best chance of understanding the impact of precarious employment experiences.

## 1. Introduction

Over the last four decades there has been a transition away from the standard employment type of full-time, permanent work with entitlements (annual/sick leave, etc.) to less standard or more flexible terms of employment, with increased casualization of the workforce, fixed-term contracts and part-time work, low wages, limited workplace rights and protection and individual-level bargaining over employment conditions [1]. The term precarious employment has increasingly been used to describe this less stable, and often detrimental, employment form [2]. However, the definition of precarious employment is broader than the contractual features of precarious employment and also encompasses the social aspects of precarious employment relationships and, specifically, workplace power relationships, e.g., powerlessness to exercise workplace rights or helplessness against workplace authoritarianism [1,3,4]. Precarious employment working arrangements may apply to all workers and not just those who work in less secure employment [5].

Considerable work has been done that shows that precarious employment impacts adversely on the mental and physical health of exposed workers [6,7,8], and on work-related injuries [9]. Employment for women has been shown consistently to be more precarious than that of men [10]. Similarly, precarious employment in migrant workers is higher than in native-born workers [11,12]. However, little is known about the impact precarious employment has on the health of migrant women [13] or if the impact differs from migrant male workers [1]. The intersections of gender and class or gender and migration status, for example, produce an experience of work and its arrangements that might vary qualitatively and structurally between women and men and between migrants and native-born workers [14]. In addition, measures of precarious employment have revealed more than one dimension related to that concept. They include vulnerability, temporariness and the ability to exercise workplace rights [4]. It is unclear how the different dimensions of precarious employment adversely impact health, or if some dimensions are more health damaging than others [1]. A review of studies found that the anticipation of losing a job or feeling that the job may be temporary were most closely associated with increases in risk factors for adverse health as well as actual adverse changes in health [15].

External migration is a major global phenomenon involving the movement of people and families between countries. In 2017, there were an estimated 258 million external migrants worldwide. Of those, 164 million were estimated to be working [16]. Australia’s mix of migrant intake has mirrored the country’s industrial and economic development. In the post-war period up until the 1970s, nearly six million European migrants arrived to build Australia’s manufacturing industries and infrastructure. These skilled, semi-skilled and unskilled workers obtained largely secure employment and migrated permanently. As the country started to transition in the mid-1970s from a manufacturing and industrial to a service-based economy, migrants with higher skill and educational levels were sought. Migrants from Southeast Asia began to arrive at this time, and it became increasingly difficult to come to Australia as an unskilled worker [17]. Since this time, there has been restructuring and deregulation of the labour market, and Australian manufacturing has declined and has moved offshore. In the 1980s, new migrants largely entered precarious employment (casual and fixed-term contract work) and temporary migration was encouraged [18]. The requirement for skilled workers increased further in the mid-1990s and is now the largest migration stream. Today, the majority of new migrants come from more than 180 countries and make up 29.7% of the Australian population and, at the latest census, make up 35% of the Australian workforce [19]. Four of the top five countries contributing to the migrant workforce are China (2.7%), India (2.6%), New Zealand (2.2%) and the Philippines (1.2%) [20,21]. Despite these large numbers, the prevalence of precarious employment or its health impacts among migrant workers is relatively unexplored, particularly with regard to differences between men and women. The aims of this present study were to investigate the two dimensions of precarious employment which have been suggested as the most important in terms of the impact on health, namely, perceptions of vulnerability and insecurity in current employment [15]. We investigated (1) if the prevalence of vulnerability and insecurity differ by migration status and gender; (2) the impact of vulnerability and insecurity on the self-reported health of women compared with men and migrants compared with Caucasian Australians, and (3) the relative impact of vulnerability and insecurity on health.

## 2. Materials and Methods 

This study used data from two cross-sectional telephone surveys investigating working conditions among Australian workers. The first was conducted in 2017 with Australian-born workers of Caucasian ancestry and the second in 2018 with workers in Australia born in New Zealand, India or the Philippines. Full details of the sampling have been described in a previous study [22]. Briefly, the first survey, conducted in 2017, recruited a sample of Australian-born workers of Caucasian ancestry aged 18 and over stratified by state and area of residence. Phone numbers were obtained from the most recent version of the Electronic White Pages (EWP), which included both landline and mobile telephone numbers. The second survey recruited workers born in New Zealand, India or the Philippines. Four strategies were needed to obtain sufficient samples for each group. The first two strategies used random sampling of the latest EWP, stratified by state and area of residence and filtered by the most common surnames for peoples born in the target countries. Further refinement was made some months into the survey to include only the suburbs that had the highest proportion of the residents in the target migrant groups, as identified by the latest census. A list of migrant contacts bought from a sample broker and recruitment through advertising and a website were the additional two recruitment strategies adopted. These four strategies successfully recruited enough participants from migrant worker groups to enable investigation of a range of working conditions.

Migrant workers were asked if they were happy to participate and were also offered the option of completing the interview in Hindi or Tagalog. Information was collected on sociodemographics and employment conditions. The Australian and New Zealand Standard Classification of Occupations (ANZSCO) was used to code occupation [23]. Ethics approval for both surveys was obtained from the Human Research Ethics Committee of Curtin University (HREC RDHS-55-16). 

### 2.1. Measures of Employment Precariousness

The two dimensions of precariousness chosen for this study were: vulnerability (which includes the latent perception of defencelessness to authoritarian treatment) and job insecurity (which includes the latent perception of job temporariness). Factor analysis of nine questions about perceptions of employment conditions confirmed a two-factor structure aligning with previous precarious employment dimensions, vulnerability and insecurity. The scree plot identified two factors with an Eigenvalue above 1 and the factor rotation matrix showed each factor above 0.5 (data not shown). The first factor, perceived vulnerability in current employment, comprised five questions (easily replaced, afraid of being fired, treated unfairly, feeling unsafe and feeling defenseless) taken from the Employment Precariousness Scale (EPRES) where they were also a factor for vulnerability [4]. Each question had three response options, agree (3), unsure (2) and disagree (1). We used these questions in two ways; first we summed them to provide a vulnerability scale with high scores indicating greater vulnerability, which provided maximum power for regression analysis. To compare with work done previously [24], we also divided the questions into three levels of vulnerability: none (no indicators of vulnerability, all questions rated unsure or disagree); low to moderate (one or two indicators of vulnerability, one or two questions rated agree) and high (three or more questions rated agree). The second factor, insecurity, comprised four questions. Three questions asked about perceptions of security in current employment (confidence the company would still be in business in five years; feel secure in current job and worry about the future of current job, came from a validated measure of job quality [25]. Responses to these three questions were rated on a Likert scale from 1 = strongly agree to 7 = strongly disagree (the question on confidence the company would still be here in five years was reverse-scored). The fourth question was based on an EPRES factor called temporariness of employment [4]. Two questions assessed preference for type of contract (casual, fixed-term, permanent) comparing present type with preferred type. The two questions were combined to assess preference from more to less secure employment (scored 0); preference for no change to existing contract (scored 1); and preference from less secure existing contract to a more secure contract (scored 2). Again, we used this factor in two ways, first we summed the questions to form a job insecurity scale with high scores indicating greater insecurity. Then, we divided the levels of insecurity using lowest and highest quartiles: low (0–6); moderate (7–11) and high (12 or higher), as done in the job quality survey [25,26].

### 2.2. Indicators of Health

The indicators of health were based on three self-reported measures. The first two were ratings of current health taken from the Medical Outcomes Study Short-Form 36 scale [27] and validated as measures of general health [28]. On a Likert scale, participants were asked to rate their current health (SF1) from excellent (1) to poor (5) and then to rate their health compared with the previous year (SF9) from much better (1) to much worse (5). The two Likert scales were tested for proportional odds in order to conduct ordinal regression but did not meet the criteria, so they were dichotomized into: General health, good to excellent (0), fair to poor (1); Health compared to last year, no change to much better (0) versus worse to much worse (1). The third measure was the Kessler 6 (K6), a validated measure of anxiety and depression [29]. This was dichotomized into K6 scores below 19 indicating less than high levels of psychological distress (0) versus scores at or above 19 (1), indicating high to very high levels of psychological distress. These cut points were developed for the Australian population [30]. As we were interested in the relative importance of vulnerability and job insecurity on well-being, we used principal component analysis on the SF1, SF9 and K6 measures, which identified a single component (Appendix A). We were therefore able to sum the three measures of physical and mental health indicators to make up the indicator of health (range 3–40). Higher scores indicated poorer health. Tests showed that the scores were not normally distributed, but no transformations were adequate.

### 2.3. Analysis

Data were weighted using Iterative Proportional Fraction [31] by age, gender, area of residence and education for workers within each population group using proportions from the 2016 Australian census [19]. Weighted univariate statistics produced population estimates with 95% confidence limits for sociodemographics, employment variables and levels of vulnerability and insecurity by gender and country of birth. Statistically significant differences were indicated by confidence limits, Chi-square tests or *t*-tests, as appropriate. Missing values were tested for missing completely at random [32]. Logistic regression modelled the associations between levels of vulnerability and insecurity with the SF1 the SF9 (physical health) and the K6 (psychological distress). Negative binomial regression modelled the association between the overall health indicator (combined physical and mental health scores) with vulnerability and insecurity entered as continuous variables (and the interaction term explored). Covariates for all regression models were sex, country of birth, age, education, area of residence, years resident in Australia (migrants) employment status, employment type, hours of work, number in company and occupation. Other than sex and country of birth, final models retained only statistically significant covariates [33], with backward deletion using one variable at a time. Variables for deletion were determined using post-estimation tests of association predictive margins were used to provide adjusted mean scores for vulnerability and insecurity levels by gender and country [34]. Final regression models were bootstrapped with bias-corrected confidence intervals [35] and post-estimation tests conducted [34]. 

## 3. Results

For the survey of Australian-born workers, 1217 households had someone who met the criteria and, of these, 1062 consented to participate (response rate 87.3%). Of these, 1051 identified as Caucasian Australian. For the survey of migrant workers, 2051 households contained someone who was either born in New Zealand, India or the Philippines, currently working and aged over 18 years. Of this total, 1630 agreed to participate (response rate 79.5%). All missing values except occupational status were missing completely at random. There were 271 (10.1%) missing values for occupational status, all Australian-born. As the percentage missing was borderline for recommended imputation (10%) and as occupational status was not statistically significantly related to health, no imputation was conducted.

Men worked more hours than women, independent of migrant status and were more likely to be self-employed or work full time (Table 1). More women born in New Zealand were recruited in comparison to men, and more men born in India were recruited in comparison to women. Twice as many men as women were employed as machinery operators or labourers, and New Zealanders had lived in Australia longer than the other migrant groups. 

Overall, women were more statistically significantly likely to experience vulnerability at work than men (Table 2). When the mean scale scores for vulnerability were compared between women and men, women had higher mean vulnerability than men (µ = 6.5 vs. µ = 5.5, t = 5.40 *p* < 0.0001, data not shown). For insecurity, more women reported low insecurity at work than men, and more men reported moderate insecurity. When the mean scale scores for insecurity were compared between women and men, men had higher mean insecurity scores than women (men µ = 9.3 vs. women µ = 8.6 t = −4.160 *p* < 0.0003, data not shown).

Among women, patterns of vulnerability and insecurity were similar across the country of birth groups. Among men, workers from India had a higher prevalence of high insecurity than workers from Australia, New Zealand or the Philippines. Patterns of vulnerability and insecurity were similar between men and women across the country of birth groups, except that fewer Australian-born women reported no vulnerability at work compared with Australian-born men. 

Over women and men combined, workers born in the Philippines had a higher mean vulnerability score compared with other migrant workers (µ = 6.5 vs. µ = 5.8 t = −3.47 *p* < 0.0003, data not shown), whereas workers born in India had a higher mean insecurity score compared with other migrant workers (µ = 9.8 vs. µ = 8.9, t = −6.1 *p* < 0.0001, data not shown).

Increasingly poor health, both physical and mental, was linearly associated with vulnerability and job insecurity except for psychological distress, where the odds of reporting high to very high psychological distress was marginally higher for the moderate job insecurity group than for the high job insecurity group (Table 3). Reporting high to very high psychological distress was six times as likely for those who also reported high vulnerability at work. Workers reporting high job insecurity were approximately three times more likely to report fair to poor health, health worse or much worse than last year and high to very high psychological distress than workers who reported low job insecurity. Similarly, workers in moderately insecure jobs were more likely to report adverse health outcomes compared with workers in jobs with low insecurity.

Health measures did not differ significantly between any of the foreign-born groups, but men were 24% less likely to report fair-to-poor health compared with women (Table 3).

Using the health indicator score as a proxy measure of overall health and well-being, associations with vulnerability and well-being showed that workers born in Australia had statistically significant poorer health compared with workers born outside Australia, for all levels of vulnerability and job insecurity. Health did not differ significantly between any of the foreign-born groups (Table 4). 

Vulnerability had the highest impact on health (Coef.0.34 *p* < 0.0001) compared with insecurity (Coef.0.25 *p* < 0.0001), but there was no significant interaction between vulnerability and insecurity (Coef.0.01 *p* = 0.114) (data not shown). 

As it is possible to feel both vulnerable and insecure, the combinations of vulnerability and insecurity were explored. The greatest impact on health was the combination of high vulnerability and high insecurity (Coef.2.37 *p* = 0.002), followed by high vulnerability and moderate insecurity (Coef.2.0 *p* = 0.007) (data not shown).

## 4. Discussion

This study examining factors of precarious work showed that women were more likely to report vulnerability at work, whereas men were less likely to report working in jobs with low insecurity and more likely to report working in jobs with moderate insecurity than women. Men from India were more likely to report working in jobs with high insecurity compared with Australian-born men. The impact of vulnerability and job insecurity on health varied by country of birth, but not gender. Compared with workers born elsewhere, workers born in Australia showed a consistent trend toward poorer health when physical and mental health indictors were combined. In this study, vulnerability was more health damaging than insecurity. 

The gender differences in our results generally concur with other work. We found that the prevalence of low/moderate insecurity was higher in men than women, but there was no difference by gender in the prevalence of high insecurity. Vulnerability was higher in women than men. Work from Spain examining employment precariousness reported a higher prevalence of precariousness in women than men (52.9% vs. 43%, *p* < 0.001) [24]. In Australia, precarious working conditions, measured as an index, have increased from 2009 for both men and women, but more rapidly for men than women; however, the prevalence remains higher in women [36]. When looking at the separate components of precariousness, the pattern is not always the same. For example, a systematic review of the literature examining the distribution of employment conditions in women and men found a higher prevalence of high job insecurity in women than men [10]. 

In our study, differences in the percentage reporting job insecurity existed between migrants and Australian-born workers. Males from the Philippines and India had a higher prevalence of insecurity than Australian-born males. This finding is similar to work from Spain that found that migrant workers had a higher prevalence of precariousness at work than native-born workers (77.1 vs. 45.2%, *p* < 0.001), particularly among young, female immigrant workers (88.6%) [24]. Similarly, a lower prevalence of US-born agricultural workers reported insecure work compared with foreign-born workers [37]. 

A review of the literature examining the impact of employment insecurity on health over the period 1980–2006 found no difference in the impact of employment insecurity on either psychological or physical health by gender [38]. However, a more recent review found moderate quality evidence of an adverse effect of employment insecurity (OR 1.52, 95% CI 1.35–1.70) on mental health [39]. This supports findings from Spain, where population-attributable risks (PAR) for mental ill health were higher among females and immigrant workers compared with their counterparts. The PAR was 7.7 for non-manual Spanish and 14.7 for immigrant men, while for non-manual Spanish and immigrant women it was 20.0 and 34.3, respectively. The pattern was similar, but more marked, for manual workers [24]. Our study found no difference in the impact of either vulnerability or insecurity on the overall health indicator (physical plus mental) health of women and men. However, the impact of vulnerability and insecurity on the overall health indicator did differ by migration status, with Australian-born workers reporting poorer self-reported health than workers born elsewhere. 

In our study, the perception of high vulnerability at work was associated with higher psychological distress than the perception of high insecurity (OR 6.24 compared with 3.11, Table 3). These results are in line with a study of the working population of Spain, aged 16–65, which found that vulnerability, as measured by the EPRES scale, correlated most strongly with perceived poorer health, particularly mental health (correlation coefficient −0.343, *p* < 0.01). Furthermore, those authors found that personality characteristics which favour answering questions negatively did not influence their vulnerability items [4]. However, our study showed that when physical health and mental health measures were combined to produce a proxy of overall health, the impact on health appeared to be greater on the Australian-born worker than the worker born overseas. This is not a finding that has been presented before to our knowledge, and it indicates that we might need to take a more wholistic approach to the impact of precariousness in the workplace, as has been proposed by Benach [1].

A possible explanation for the greater health impact in our study of vulnerability than insecurity and for the differential health impact between workers born in Australia compared with workers born in India or the Philippines is that both India and the Philippines have large informal labour markets where precarious work conditions, such as unstable and insecure work practices, are the norm. Additionally, the formal labour market in these countries is undergoing a rapid process of informalisation, at the same time that the informal labour market is growing in size [40]. The experience of these workers in their home countries may have been working in the informal labour market, thus making them more familiar with precarious working conditions and particularly insecurity. We did not ask about working conditions in the home country, so we are unable to assess this in this current study. 

Finally, there appears to be a gap in the literature concerning the place of precariousness in stress related to the workplace. While there are some theoretical models of the role of stress in the workplace, they are almost exclusively based on the concept of job satisfaction focusing on concepts such as the level of demand and control within the workplace [41]. Some recent work has examined the role of ethnicity in coping with stress in the workplace [42]. However, the role of precariousness, which we would argue is as prevalent and as important a source of stress, has not been discussed in these works. There would appear to be a place for some investigation into how ethnicity with its concurrent cultural mores affects the perception of precarity and stress in the workplace. Our descriptive study suggests that some cultural mores may have an ameliorating role, at least in Australia.

## 5. Strengths/Limitations

As the study was a self-reported cross-sectional survey, no attribution of causality can be made. Telephone surveys can be influenced by a number of factors including, in the case of migrant workers, fear of consequences for participating [43] and fear of being identified which might impact their work [44]. However, the high participation rate (79.4%) suggests that few of the workers we contacted were in these categories. A further limitation of this study was the need to include two non-probability sampling strategies to achieve the required number of workers within each migrant group. Weighting the data to the census for workers from each migrant group helped to address any bias in population estimates and there was an unexpected bonus. The sample we used from the sample broker, who only provided mobile numbers, yielded more workers who were young, male and from lower socioeconomic levels. This supports previous research, which argues for the importance of using flexible methods when researching “hard-to-reach” populations [45].

We only looked at two dimensions of precarious employment and there are more as identified by Vives [4], and there may be other dimensions that impact negatively on health to a greater or lesser extent.

A strength of this study is that we collected quantitative individual-level data from workers across the population, rather than just in high-risk industries. This allowed us to include all job types allowing for the results to be generalised to the population, which is unusual in studies on migrant workers.

## 6. Conclusions

Perceptions of vulnerability and insecurity in current employment, dimensions of precarious employment, were strongly associated with poorer self-reported health. Vulnerability had a greater negative impact on health status than insecurity. Women and Caucasian Australians reported the poorest health for both vulnerability and insecurity. These results suggest that understanding both changes in employment conditions over time as well as knowledge of cultural patterns probably offer the best chance of assessing the impact of precarious work experiences.

## Figures and Tables

**Table 1 ijerph-17-07540-t001:** Unweighted number and weighted prevalence estimates with 95% Confidence Intervals ^a^ for participant sociodemographics and employment status by gender.

Participant Characteristics	Females (*n* = 1374)	Weighted % (95% CI)	Males (*n* = 1307)	Weighted % (95% CI)
Country of birth				
Australia	606	42.6 (39, 46.2)	445	35.8 (32.2, 39.6)
New Zealand	328	23.5 (20.7, 26.5)	238	16.9 (14.6, 19.5)
India	231	13.8 (11.6, 16.4)	402	27 (23.8, 30.4)
Philippines	209	20.2 (17.5, 23.2)	222	20.3 (17.6, 23.2)
Age group ^b^				
18–45 years	539	64.6 (61.6, 67.5)	601	68.6 (65.6, 71.5)
46–65 years	830	35.4 (32.5, 38.4)	701	31.4 (28.5, 34.4)
Missing	5		5	
Highest level of education ^b^				
School only	265	30.4 (26.8, 34.2)	201	29.7 (25.8, 33.8)
Trade/Diploma/Certificate	415	23.1 (20.5, 25.9)	436	25.9 (23.1, 28.9)
Tertiary	701	46.5 (43, 50.1)	669	44.4 (40.9, 48)
Missing	2		1	
Area of residence				
Major metropolitan	999	74.8 (71.6, 77.6)	974	78.2 (74.9, 81.1)
Rest of state	384	25.2 (22.4, 28.4)	335	21.8 (18.9, 25.1)
Employment status				
Self-employed	153	9.8 (8, 11.9)	256	18.1 (15.3, 21.3)
Work for others part time	577	45.3 (41.7, 48.9)	174	18.1 (15.2, 21.6)
Work for others full time	644	45 (41.5, 48.6)	876	63.4 (59.6, 67.1)
Missing	0	0	1	0.3 (0, 2.2)
Employment type				
Casual	246	22.4 (19.4, 25.8)	143	15.2 (12.5, 18.3)
Fixed-Term Contract	158	11.4 (9.4, 13.8)	133	10 (7.8, 12.7)
Permanent	970	66.1 (62.6, 69.5)	1031	74.8 (71.3, 78.1)
Number working in company				
Up to 19 workers	343	23.9 (21, 27)	430	34.9 (31.4, 38.7)
20–199 workers	340	25.7 (22.7, 29)	288	23.5 (20.4, 26.8)
200 & over workers	680	49.4 (45.8, 53)	574	40.5 (37, 44.1)
Missing	11	1 (0.5, 1.9)	15	1.1 (0.6, 2.1)
Occupation				
Managers/Professionals	465	28.1 (25.2, 31.1)	466	29.7 (26.7, 33)
Technician/community services/clerical/sales	645	50 (46.5, 53.6)	505	38.9 (35.3, 42.5)
Machinery operators/Labourer	112	11.4 (9.1, 14.1)	217	21.2 (18, 24.9)
Missing	152	10.5 (8.4, 13)	119	10.2 (8, 12.9)
Mean hours worked weekly				
Australia	606	30.2 (28.6, 31.7)	444	39.4 (37.1, 41.6)
New Zealand	328	33.6 (31.7, 35.6)	238	43.1 (40.6, 45.6)
India	230	30.6 (28.6, 32.7)	400	37.6 (35.5, 39.7)
Philippines	208	30.5 (28.6, 32.3)	220	38.5 (36.6, 40.5)
Mean years in Australia *				
New Zealand	328	19.5 (18.1, 20.9)	236	19.4 (17.9, 21)
India	229	13.1 (11.7, 14.5)	398	12.2 (11.2, 13.1)
Philippines	209	14.6 (13.1, 16)	220	15.3 (13.3, 17.3)

Note: ^a^—The confidence intervals were used to indicate significant differences between the genders (italicised, bolded numbers on the table); ^b^—Age and education were weighting variables so there were no weighted estimates for missing values. 95%CI—95% Confidence Interval. *—Mean years in Australia only apply to migrant workers.

**Table 2 ijerph-17-07540-t002:** Weighted prevalence estimates of levels of vulnerability and job insecurity by country of birth group for women and men.

xxx	Vulnerability			Job Insecurity		
	None	Low/Moderate	High	Low Insecurity	Moderate Insecurity	High Insecurity
	% (95% CI)	% (95% CI)	% (95% CI)	% (95% CI)	% (95% CI)	% (95% CI)
**Women**	**56.7 (53.1, 60.2)**	27.2 (24.2, 30.6)	16.1 (13.5, 19.0)	54.3 (50.6, 58.0)	27.0 (23.9, 30.3)	18.7 (15.9, 21.9)
Country of birth						
Australia	**56.7 (50.9, 62.4)**	26.7 (21.8, 32.4)	16.5 (12.5, 21.5)	53.0 (46.9, 58.9)	28.4 (23.4, 34.1)	18.6 (14.0, 24.3)
New Zealand	63.0 (56.1, 69.4)	25.2 (19.4, 32.0)	11.8 (8.5, 16.3]	56.8 (49.7, 63.7)	24.4 (19.1, 30.6)	18.8 (13.5, 25.4)
India	52.0 (42.7, 61.1)	30.0 (22.5, 38.9)	18.0 (10.9, 28.3)	48.8 (39.1, 58.5)	30.2 (22.4, 39.2)	21.1 (14.5, 29.6)
Philippines	51.3 (43.4, 59.2)	29.7 (23.3, 37.0)	19.0 (13.3, 26.4)	57.6 (49.5, 65.4)	24.7 (18.8, 31.9)	17.6 (12.3, 24.6)
Mean years in Australia #	17.0 (15.8, 18.2)	14.0 (12.5, 15.4)	17.4 (15.3, 19.5)	16.6 (15.4, 17.8)	16.3 (14.5, 18.1)	14.2 (12.4, 15.9)
Men	**66.5 (63.0, 69.8)**	22.3 (19.5, 25.5)	11.2 (9.2, 13.6]	**46.8 (43.0, 50.6)**	**35.2 (31.6, 39.0)**	18.0 (15.4, 21.0)
Country of birth						
Australia	**71.0 (64.4, 76.9)**	18.3 (13.4, 24.4)	10.7 (7.2, 15.7)	50.1 (43.0, 57.2)	37.5 (30.7, 44.8)	12.4 (9.0, 16.9)
New Zealand	65.5 (57.9, 72.4)	24.2 (18.2, 31.3)	10.3 (6.5, 16.1)	51.1 (43.1, 59.1)	34.3 (27.2, 42.3)	14.5 (10.1, 20.5)
India	67.2 (60.8, 72.9)	20.1 (15.5, 25.6)	12.8 (9.4, 17.2)	41.6 (34.9, 48.6)	34.3 (28.1, 41.1)	24.1 (18.4, 31.0)
Philippines	58.2 (50.6, 65.4)	31.1 (24.6, 38.4)	10.7 (7.1, 16.0)	44.5 (36.7, 52.5)	33.6 (26.5, 41.5)	22.0 (16.1, 29.2)
Mean years in Australia #	15.9 (14.8, 17.0)	13.2 (11.8, 14.6)	14.6 (11.8, 17.4)	15.4 (14.2, 16.6)	14.8 (13.2, 16.5)	14.5 (12.5, 16.5)

Note: Bolded figures show confidence intervals that are not overlapping on the same variables between men and women indicating statistically significant differences; bolded, italicised figures show confidence intervals that are not overlapping on the same variables across country of birth groups, indicating statistically significant differences; #—the mean years in Australia only apply to migrant workers.

**Table 3 ijerph-17-07540-t003:** Adjusted Odds Ratios and 95% Confidence Intervals for health outcomes for levels of vulnerability, job insecurity, gender and country of birth.

Characteristics	SF1 (Fair to Poor Health)		SF9 (Health Worse or Much Worse Than Last Year)		K6 (High to Very High Psychological Distress	
	aOR (95% CI)	*p*	aOR (95% CI)	*p*	aOR (95% CI)	*p*
No vulnerability	1 (Reference)		1 (Reference)		1 (Reference)	
Low–moderate vulnerability	1.49 (1.11, 1.99)	0.006	1.39 (1.05, 1.86)	0.022	1.65 (0.75, 3.65)	0.228
High Vulnerability	1.81 (1.32, 2.49)	<0.0001	2.15 (1.59, 2.8)	<0.0001	6.24 (3.46, 12.23)	<0.0001
Low job insecurity	1 (Reference)		1 (Reference)		1 (Reference)	
Moderate job insecurity	1.54 (1.17, 2.18)	0.005	1.65 (1.28, 2.25)	<0.0001	3.46 (1.75, 7.63)	0.001
High job insecurity	2.91 (2.09, 4.07)	<0.0001	2.96 (2.085, 4.04)	<0.0001	3.11 (1.54, 6.74)	0.004
Female	1 (Reference)		1 (Reference)		1 (Reference)	
Male	0.76 (0.59, 0.98)	0.037	0.86 (0.67, 1.09)	0.210	0.72 (0.4, 1.13)	0.218
Born in Australia	1 (Reference		1 (Reference)		1 (Reference)	
Born in New Zealand	1.19 (0.84, 1.61)	0.273	1.14 (0.8, 1.53)	0.408	0.8 (0.39, 1.77)	0.579
Born in India	0.74 (0.51, 1.04)	0.099	0.98 (0.72, 1.3)	0.889	0.95 (0.47, 1.63)	0.871
Born in the Philippines	0.79 (0.53, 1.147)	0.219	0.83 (0.58, 1.22)	0.346	0.46 (0.14, 1.13)	0.162

Note: Covariates entered initially: sex, cob (retained in all models), age, area of residence, time in resident in Australia (migrants), employment type (self vs. works for others) and present contract (casual, fixed-term or permanent), number in the company, hours working weekly and occupational group; covariates retained in the final model: sex, country of birth and the only remaining statistically significant employment variable—present contract (casual, fixed-term or permanent); aOR—adjusted odds ratio; 95%CI—95% Confidence Interval; SF1—Short Form question 1 about current health; SF9—Short Form question 9 comparing health with previous year; K6—Kessler 6, measure of anxiety and depression.

**Table 4 ijerph-17-07540-t004:** Estimated mean health-indicator scores for gender and country of birth by levels of vulnerability and job insecurity, using predictive margins based on the regression equation.

**Characteristics**	**Adjusted ^a^ Mean Health-Indicator Score ^b^**		
	**No Vulnerability Mean (95% CI)**	**Low–Moderate Vulnerability (95% CI)**	**High Vulnerability (95% CI)**
Male	13.5 (13.2, 13.7)	14.6 (14.4, 14.9)	16.8 (16.3, 17.3)
Female	13.8 (13.5, 14.0)	14.9 (14.7, 15.2)	17.2 (16.7, 17.7)
Born in Australia	**14.2 (13.9, 14.5)**	**15.5 (15.2, 15.8)**	**17.9 (17.3, 18.4)**
Born in New Zealand	13.5 (13.2, 13.8)	14.71 (14.4, 15.1)	16.9 (16.3, 17.5)
Born in India	13.1 (12.8, 13.4)	14.4 (14.0, 14.7)	16.5 (15.9, 17.1)
Born in the Philippines	12.9 (12.6, 13.3)	14.1 (13.7, 14.5)	16.3 (15.7, 16.9)
	**Low Job Insecurity (95%CI)**	**Moderate Job Insecurity (95%CI)**	**High Job Insecurity (95% CI)**
Male	13.4 (13.1, 13.6)	14.5 (14.3, 14.8)	16.1 (15.7, 16.5)
Female	13.7 (13.4, 13.9)	14.9 (14.6, 15.1)	16.5 (16.0, 16.9)
Born in Australia	**14.1 (13.9, 14.4)**	**15.4 (15.1, 15.7)**	**17.1 (16.6, 17.5)**
Born in New Zealand	13.4 (13.1, 13.7)	14.6 (14.2, 14.9)	16.2 (15.7, 16.7)
Born in India	13.0 (12.7, 13.4)	14.2 (13.9, 14.5)	15.8 (15.3, 16.2)
Born in the Philippines	12.8 (12.5, 13.2)	13.9 (13.6, 14.4)	15.5 (15.0, 16.1)

Note: ^a^—Estimated mean scores using predictive margins based on the negative binomial regression equation adjusted for sex country of birth, age, education, area of residence, employment status, employment type, number in company and occupation. The final model, used for the estimates, only retained the statistically significant associations (Appendix A shows the final model). Bolded numbers indicate that confidence intervals are outside of the range of workers born overseas; ^b^—Higher mean health-indicator score indicates poorer self-reported health; 95%CI—95% Confidence Interval

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
