# Peer review of "Examining the Impact of Two Dimensions of Precarious Employment, Vulnerability and Insecurity on the Self-Reported Health of Men, Women and Migrants in Australia"

_ijerph, 2020, doi:10.3390/ijerph17207540_

Round 1

Reviewer 1 Report

Dear Authors,

I find it an interesting article that provides the literature with empirical evidence of the impact that precarious employment has on the health of migrant women and men living in Australia, especially in India and the Philippines.

The evaluation scales used are adequate, both the ESPRES and the MOS have been validated and have significant reliability in the literature.

It is a good job that contributes to the generation of knowledge about the employment situation of workers.

Reviewer 2 Report

The paper examines the effect of employment conditions (i.e. precarious employment conditions) upon self-reported health, among two large samples of workers in Australia from different ethnic backgrounds. This is a very interesting and well-written paper, focusing on a popular topic of employment conditions and their relationship to health. I don't have any major comments, except that in the Limitations part, you should add potential endogeneity issues that have not beed addressed and that they might affect the estimates of interest. In what way do you address the issue of endogoneity/simultaneity in the employment-health relationship? I believe you should at least discuss this source of bias on the Limitations section.

Reviewer 3 Report

Introduction A poor contextualization of migratory problems in the world is observed, and which of these currents affect the study subjects. The introduction should refer to the migratory movements that Australia has suffered in recent decades, both internally and externally. I would like the authors to respond to me because migratory movements of European citizens and Middle Eastern countries have not been taken into account. Material and method 73 Philippines. Full details of the sampling have been described in a previous study. If a previous article is published, it should be explained what is new about the current one, since they are data from more than two years. Results Workers born in Australia had statistically significant poorer health compared with workers born 187 in the Philippines, for all levels of vulnerability and job insecurity and statistically significantly poorer 188 health than workers born in India for all levels of insecurity. Health did not differ significantly 189 between any of the foreign-born groups (Table 3). These data should be explained more in depth as they contradict the bibliography published by the UN and UNHCR. They should indicate how they have checked that the subjects were not subject to bias when collecting the data. A table by production sectors would be necessary since not all jobs are affected by the same factors when carrying out workloads. Discussion I recommend that you review the immigration data from both Europe and the United States as they are updated and there are appointments from 2019, You should look for more quotes that support your conclusions and support your hypothesis, and the quality of them. Bibliography 18/37 citations are more than five years old. The most recent literature and studies should be offered.

Author Response

Please see attached file for our response to your comments

Reviewer 4 Report

The aims and methods of the paper were presented well, and the results are interesting but they mostly spark interest as opposed to offering policy relevant conclusions.

First, as the authors themselves suggest, causality is going in multiple directions so it is unclear as the underlying structure that is causing the observed differences. This should be discussed further in regards to interpreting the results.

Second, unless I am misunderstanding the methodology, there are no controls for things like education, age, type of occupations, etc. even though those variables have been collected and reported in describing the sample. If differences in vulnerability and insecurity went away once these were controlled for we could then locate the root causes in pre-market conditions that lead to the type of jobs that people are able to obtain. If not, then there is something happening on the job.  Being able to apportion differences between these two things would have great relevance for what sorts of policies would be best suited to equalizing outcomes.

If I am wrong about this...please make this more explicit. But if I am right, I would want to see this type of analysis done.

As it stands now the measured differences are important and make me want to know more, but don't give me much guidance on what to do with them.

I would recommend publication if these issues could be addressed in a way that point to potential policy interventions.

I hope the authors can address this,.

Author Response

(The authors gave the same response as above.)

Reviewer 5 Report

The topic concerning precarious employment is worthy to explore but the method approach adopted are not appropriate.

Major concerns:

Method flaw: The authors selected items of some validated scales collected from the survey and perform factor analysis to regenerate new scales to measure both the concept of precarious employment and health subsequently. The detailed factor analysis should be reported and cross-validated against other scale.

Measurement of precarious employment: In the introduction the authors briefly described the precarious employment as detrimental and less stable employment form and extend it to the workplace power relationship including powerlessness to exercise workplace rights or helplessness against workplace authoritarianism. However, the authors only selected two dimensions (vulnerability and insecurity) from the Employment Precariousness Scale (EPRES) to measure the precarious employment. The items selected do not measure any workplace power relationship. The survey may not collect data using all items from the EPRES, but the authors should explain and provide evidence on how these two dimensions related to workplace power relationship or how these two dimensions greatly reflect the concept of precarious employment.

Measurement of health: The authors combined all three validated scales into one measurement by claiming a one factor dimension generated from the factor analysis. Please provide detailed factor analysis as appendix. The author may separate different aspect of health (mental and physical health) from one scale instead of using items from the scales and combining them.

Author Response

please see attached file for our responses to your comments

Round 2

Reviewer 3 Report

The changes suggested by this reviewer were successful.

Author Response

Dear reviewer,

Thank you for the comments.

Reviewer 4 Report

Thanks for clarifying the methodological question I had. I suspected that you did control for those other factors, but it wasn't clear to me, which is why I asked.

I do feel that the article would be stronger if you could link it to policy concerns or at least provide some direction to policy questions, even if you can't answer them.

In my experience the majority of quantitative studies of social and economic data do this.

But that said, the results here are interesting and could spark interest in that among others.

Author Response

Dear reviewer,

Thank you for the comments

Reviewer 5 Report

Although the paper has been improved by providing more background information, the methodology used for the measurement of health is not appropriate and make the significant content of the paper less important.

The author provided PCA to confirm a one factor solution of the selected items from SF36, SF9 and K6, but this doesn't mean the measurement of the full spectrum of general health.  In the discussion section, the author have showed that more recent review found moderate quality evidence of an adverse effect of employment insecurity on mental health. The authors in their study found no difference in the impact of either vulnerability or insecurity on the health of women and men. But the impact of vulnerability and insecurity on health did differ by migration status.  It seems that there is no further new insight on the impact of precarious employment on the full spectrum of health (including physical and mental health) by gender and migration status.  Additional analysis is required.

In terms of precarious employment, the authors selected two dimensions for the analysis.  As the authors claims that the items were collected from a large survey which limit them to use all dimensions of the Employment Precariousness Scale and focus only on vulnerability.  The two dimensions used is only reflect partial concept of precarious employment which should be stated in the limitation section.

The discussion section should provide more result and policy implications and highlight the new insight.

Author Response

Dear reviewer,

Please see the file attached

Round 3

Reviewer 5 Report

Additional analysis makes the manuscript more scientific soundness.  Minor spell check and reference citation style check are needed. Refinement on some sentence structure in the result section is needed for easy reading.

Author Response

Thanks, we have corrected minor typos, improved the sentence structure of a couple of sentences to increase clarity and fixed some incorrect citations. Thanks for your careful, close reading of our manuscript.